# Activation and Identification of a Griseusin Cluster in *Streptomyces* sp. CA-256286 by Employing Transcriptional Regulators and Multi-Omics Methods

**DOI:** 10.3390/molecules26216580

**Published:** 2021-10-30

**Authors:** Charlotte Beck, Tetiana Gren, Francisco Javier Ortiz-López, Tue Sparholt Jørgensen, Daniel Carretero-Molina, Jesús Martín Serrano, José R. Tormo, Daniel Oves-Costales, Eftychia E. Kontou, Omkar S. Mohite, Erik Mingyar, Evi Stegmann, Olga Genilloud, Tilmann Weber

**Affiliations:** 1The Novo Nordisk Foundation Center for Biosustainability, Technical University of Denmark, Kemitorvet, Building 220, 2800 Kongens Lyngby, Denmark; chabec@biosustain.dtu.dk (C.B.); tetgre@biosustain.dtu.dk (T.G.); tuspjo@biosustain.dtu.dk (T.S.J.); eeko@biosustain.dtu.dk (E.E.K.); omkmoh@biosustain.dtu.dk (O.S.M.); 2Fundación MEDINA, Centro de Excelencia en Investigación de Medicamentos Innovadores en Andalucía, Parque Tecnológico de Ciencias de la Salud, Av. Conocimiento, 34, 18016 Granada, Spain; javier.ortiz@medinaandalucia.es (F.J.O.-L.); daniel.carretero@medinaandalucia.es (D.C.-M.); jesus.martin@medinaandalucia.es (J.M.S.); ruben.tormo@medinaandalucia.es (J.R.T.); daniel.oves@medinaandalucia.es (D.O.-C.); 3Department of Microbial Bioactive Compounds, Interfaculty Institute of Microbiology and Infection Medicine, Eberhard Karls Universität Tübingen, Auf der Morgenstelle 28, 72076 Tübingen, Germany; erik.mingyar@gmail.com (E.M.); evi.stegmann@biotech.uni-tuebingen.de (E.S.); 4German Center for Infection Research (DZIF), Partner Site Tübingen, 72076 Tübingen, Germany

**Keywords:** transcriptional regulators, biosynthetic gene cluster, genome mining, heterologous expression, griseusin, pyranonaphtoquinone, forosamine, mycothiol

## Abstract

*Streptomyces* are well-known producers of a range of different secondary metabolites, including antibiotics and other bioactive compounds. Recently, it has been demonstrated that “silent” biosynthetic gene clusters (BGCs) can be activated by heterologously expressing transcriptional regulators from other BGCs. Here, we have activated a silent BGC in *Streptomyces* sp. CA-256286 by overexpression of a set of SARP family transcriptional regulators. The structure of the produced compound was elucidated by NMR and found to be an *N*-acetyl cysteine adduct of the pyranonaphtoquinone polyketide 3′-*O*-α-d-forosaminyl-(+)-griseusin A. Employing a combination of multi-omics and metabolic engineering techniques, we identified the responsible BGC. These methods include genome mining, proteomics and transcriptomics analyses, in combination with CRISPR induced gene inactivations and expression of the BGC in a heterologous host strain. This work demonstrates an easy-to-implement workflow of how silent BGCs can be activated, followed by the identification and characterization of the produced compound, the responsible BGC, and hints of its biosynthetic pathway.

## 1. Introduction

Bacteria of the genus *Streptomyces* are well-known producers of bioactive compounds with anti-bacterial activity. Most of these compounds are produced by large enzyme complexes, such as the polyketide synthases (PKSs) [1] and the non-ribosomal peptide synthetases (NRPSs) [2,3]. These multi-modular enzymes are encoded in specific clustered regions of the bacterial genomes, called biosynthetic gene clusters (BCGs). Recent advances in whole genome sequencing and genome mining has uncovered that the majority of BCGs are not expressed under standard laboratory conditions, and are thus called silent [4,5]. Several methods for the activation of silent BGCs exist, including heterologous expression, promoter engineering, ribosome engineering, and engineering of transcriptional regulators [6]. Recently, a “semi-targeted” approach of overexpressing heterologous regulators in *Streptomyces* resulted in the activation of previously silent BGCs of bioactive compounds [7]. Overexpression of transcriptional regulators in other hosts allows for high throughput screening of isolates, and is the method used in this study. One class of transcriptional regulators, that are frequently used for similar purposes, is the *Streptomyces* antibiotic regulatory protein (SARP) family regulators. They are pathway-specific transcriptional regulators that are only found in actinomycetes and recognize a part of the promoter sequence of the gene cluster that they regulate [8]. It has previously been shown that overexpressing a SARP family regulator can activate production from silent BGCs in *Streptomyces*. In one study, the SARP gene *papR2* from *Streptomyces pristinaespiralis* was overexpressed in *Streptomyces lividans* resulting in activation of the silent undecylprodigiosin (Red) BGC [9]. These types of transcriptional activations of silent BGCs by SARP regulators are only possible because many BGCs are associated with highly similar SARP regulators. In this study, four plasmids encoding different classes of transcriptional regulators from *Streptomyces* were used (Table 1). They are all based on the integrative plasmid pRM4 [10] and the transcriptional regulator genes are under control of the constitutive promoter P*ermE** [11]. The four plasmids encode: cluster specific regulators (CSRs) containing the four genes *aur1P-pntR-strR-fkbN*; *Streptomyces* antibiotic regulatory proteins (SARPs) containing the five genes *actIIORF4-griR-aur1PR3-papR2-redD*; gamma butyrolactone synthases (GBLs) containing the two genes *scbA-afsA*; and global regulators containing the five genes *afsR-adpA-crp-absB-dasR*. Overexpression of these different classes of transcriptional regulators are hypothesized to activate different silent BGCs. The objective of this study was to employ the four plasmids to activate production of novel compounds in a collection of actinomycete strains isolated from soil samples. A combination of multi-omics methods, gene inactivations, and heterologous expression was then used to identify and characterize the compound and the corresponding BGC.

## 2. Results and Discussion

### 2.1. Transfer of Plasmids Encoding Transcriptional Regulators, High Throughput Cultivation and Metabolomics

Four plasmids encoding different classes of transcriptional regulators (Table 1), along with the empty plasmid pRM4 [10] used as control, were transferred to a selection of actinomycetes of the MEDINA strain collection by standard intergeneric conjugation [12]. Alongside the transfer, susceptibility to apramycin and expression from the promoter P*ermE**, tested with GusA, was examined to ensure correct conjugations and expression of the regulators [13]. The strains with successfully received transcriptional regulator plasmids were cultivated in a high throughput format for identification of new compounds. The cultivations were carried out in duplicates in 10 mL of four different liquid media; FR23, DNPM, FPY12 and M016. The cultures were extracted in acetone and DMSO and analyzed with LC-MS. Activated production of new compounds in regulator-carrying strains were identified using MASS Studio [14]. MASS Studio provides an overview of the abundance of each ion detected in low resolution MS. The cases where the abundance of an ion was significantly enhanced when comparing the strain with the empty plasmid pRM4 and the same strain with one of the regulator plasmids, were marked. In one strain, named CA-256286, production of a compound with an ion at *m*/*z* 749 in MO16 medium was activated by plasmid pRM4-SARPs encoding SARP family transcriptional regulators. The enhanced ion was not observed in the clean medium, in other media types or with any of the three other regulator plasmids.

### 2.2. Isolation and Structure Elucidation of Griseusin-Derived N-Acetyl Cysteine Adducts 1 and 2

HPLC-HRESIMS analysis of the culture extract of *Streptomyces* sp. CA-256286 (pRM4-SARPs) in MO16 medium allowed to assign **1** the molecular formula C_35_H_44_N_2_O_14_S, based on the [M + H]^+^ ion at *m*/*z* 749.2589 (Δ −0.40 ppm) (Appendix A). Searches in different natural products databases failed to identify any compound with the observed accurate mass, suggesting that **1** was a new natural product. The strain *Streptomyces* sp. CA-256286 (pRM4-SARPs) was then cultivated in a larger scale (MO16 medium; 2 L) and the regrowth was processed as described in the experimental section. Targeted isolation by MPLC and further semipreparative RP-HPLC (Appendix A) yielded **1** as a yellow-orange, amorphous powder. Unexpectedly, further LC-HRESIMS analysis of the peak collected as **1** revealed the presence of two species, compound **1** itself together with compound **2**, whose [M + H]^+^ ion at *m*/*z* 747.2431 indicated the loss of two hydrogen atoms compared to **1** (Appendix A).

To determine whether compound **2** was a co-eluting impurity or a product resulting from the oxidation of **1**, a sample of **1** was analyzed by LC-HRESIMS over time. After 36 h at room temperature, the **1**:**2** area ratio changed from 70:30 to 7:93, thus confirming the spontaneous conversion of the original unstable compound **1** into **2** (Appendix A). In parallel, another sample of **1** was monitored by ^1^H-NMR, and its proton signals were observed to change gradually through time until a nearly complete conversion into **2** after 48 h (Appendix A).

Considering the instability of **1** and the feasibility of structural characterization of **2**, we allowed **1** to be readily oxidized (DMSO, room temperature for 48 h) and then repurified by semi-preparative HPLC to yield **2** as an orange, amorphous powder (Appendix A). The molecular formula C_35_H_42_N_2_O_14_S was confirmed for **2** based on its HRESIMS(+)-TOF spectrum, showing [M + H]^+^ and [M + 2H]^2+^ ions at *m*/*z* 747.2434 (Δ −0.60 ppm) and *m*/*z* 374.1258 (Δ −1.60 ppm), respectively (Appendix A). Additionally, tandem-mass spectrometry of the [M + H]^+^ adduct showed a single fragment ion at *m*/*z* 142.1239, which was consistent with the presence of the monosaccharide forosamine and thus advanced the partial glycosidic nature of **2** (Appendix A).

The planar structure of **2** (Figure 1) was determined by 1D and 2D NMR spectroscopic analyses (Table 2). Interpretation of ^13^C NMR and HSQC spectra (Appendix A) revealed the presence of 13 quaternary carbons, including six carbonyl groups (among them, two quinone CO signals at ***δ***_C_ 186.3 and 181.3 ppm), one oxygenated aromatic carbon (***δ***_C_ 160.8) and two characteristic hetero atom-substituted carbons at ***δ***_C_ 95.6 and 81.9 ppm. The remaining signals were 3 aromatic/olefinic methines, 6 oxygenated methines (including one anomeric carbon at ***δ***_C_ 93.7), 3 aliphatic methines (including a characteristic α-proton of amino acid at ***δ***_C_ 51.5), 5 methylenes and 6 methyl groups. Among the latter, two singlet methyls were assigned to an *N*,*N*-dimethyl group based on their chemical equivalence in ^1^H NMR (***δ***_H_ 2.54) and ^13^C NMR chemical shift (***δ***_C_ 40.4 ppm).

From the analysis of COSY and TOCSY spectra, five spin systems were identified, as shown in Figure 2a. Three double doublets at ***δ***_H_ 7.82, 7.60 and 7.43 in the ^1^H NMR spectrum (Table 2) and their HMBC correlations (Figure 2a), jointly established a 5-hydroxy-1,4-naphtoquinone moiety (Figure 1, A and B rings). The presence of this chromophore in the structure of **2** was in good agreement with its characteristic UV spectrum (Appendix A). Two further spin systems comprising C-3/C-11 and C-3′to C-7′ together with key HMBC cross-peaks between H-3′/C-1, H-3′/C-10a and H-3/C-1 allowed us to construct the fused spiro-ring C/E system, thus illustrating that **2** was structurally related to the griseusin family of pyranonaphtoquinone antibiotics [15,16]. Within the E ring, C-4′-*O* was reasoned to be acetylated based on the downfield shift of the H-4′ proton (***δ***_H_ 5.49), so singlet methyl (***δ***_C_ 20.8; d_H_ 2.01) and carbonyl (***δ***_C_ 169.9) signals were eventually assigned to the *O*-Ac group attached to that position.

Based on the COSY/TOCSY, HSQC and HMBC spectra, another spin system comprising C-1″ to C-7″ was elucidated as a 2,3,4,6-tetradeoxy-hexose monosaccharide moiety substituted with a *N,N*-dimethyl group at C-4″. This residue was thus identified as the sugar forosamine, further supported by the ^13^C NMR fingerprint chemical shift values [17] (Table 2). The multiplicity of H-1″ as a broad singlet suggested an equatorial orientation, thus indicating an α-glycosidic bond. Mutual HMBC correlations from H-1″ to C-3′ and from H-3′ to C-1″ clearly established the *O*-linkage of the monosaccharide to C-3′ position at the ring E.

On the other hand, HMBC correlations from both H-3 and H-11 to C-4 and from H-3 to carbonyl C-12 revealed the presence of a carboxymethyl group attached to C-3. The remaining one degree of unsaturation eventually required for the molecular formula established a *γ*-lactone moiety in **2** (Figure 1, D ring), as reported for other pyranonaphtoquinones, including griseusin members of the A, F, and G series [15,18]. Remarkably, the lack of a H atom at C-4 in **2** compared to those related metabolites indicated the substitution at that position.

The analysis of the remaining 2D NMR data clarified the presence of an *N*-acetyl cysteine (*N*-AcCys) moiety in **2**. Thus, the COSY spectrum showed the connectivity of the methylene protons H-3″′ (***δ***_C_ 32.6; ***δ***_H_ 3.33, 2.94) with methine H-2″′ (***δ***_C_ 51.5; ***δ***_H_ 4.37), which in turned correlated with the NH proton (***δ***_H_ 8.22). These data, along with the additional signals derived from the acetyl (***δ***_C_ 169.3, 22.1) and the carboxyl (***δ***_C_ 171.6) groups, jointly determined the presence of the *N*-AcCys moiety. Finally, key HMBC correlations from H-3″′ methylene protons to C-4 unambiguously established the attachment of the *N*-AcCys at C-4 via the sulfur atom.

In view of the structure of **2**, it can be considered an *N*-AcCys adduct (at C-4) of 3′-*O*-α-d-forosaminyl-(+)-griseusin A (**3**), a known member of the griseusin family originally isolated from *Streptomyces griseus* [19]. Interestingly, such AcCys S-conjugates of the related pyranonaphtoquinone antibiotics kalafungin or dihydrokalafungin have been identified from recombinant *S. coelicolor* strains and their occurrence has been explained by the addition and further processing of mycothiol to those metabolites [20,21].

With regard to the relative configuration of **2**, although we could reasonably assume it to be the same as in **3**, different naturally occurring epimers at positions within the E ring have been reported for griseusins of the A and B series [22], therefore, we set out to determine it unambiguously for **2**. On the one hand, the multiplicity of H-4′ as an apparent quadruplet (q) with a small ^3^*J*_H,H_ coupling constant (3.3 Hz) indicated an equatorial orientation. On the other hand, although ^1^H-^1^H coupling constants of H-6′ could not be accurately measured, their appearance as “dqd” with at least one large coupling constant (^3^*J*_H-6′,H-5′ax_ = ca. 8–9 Hz; ^3^*J*_H-6′,H-7′_ = 6.3 Hz; ^3^*J*_H-6′,H-5′eq_ = 2–3 Hz) (Appendix A), strongly suggested an axial orientation for this hydrogen. After stereospecific assignment of methylene protons H-5′ax/eq (Appendix A), key NOESY correlations between H-3′/H-4″, H-3′/H-5′ax and H-6′/H-5′eq established the relative configuration in the pyrane E ring. Additionally, a strong *nOe* cross-peak between the distant H-7′ (axially oriented) and H-3 (D ring) completed the relative configuration assignments in **2** (Figure 2b), which were confirmed to be the same as those reported for **3**.

Although pyranonaphtoquinones, including griseusins, may exist in two enantiomeric versions [15,23,24,25], the 3′-*O*-α-(d)-forosaminyl derivative (**3**) has only been found associated with (+)-griseusin A [19], which in turned was shown to be the enantiomer of the formerly isolated griseusin A [26,27]. Considering the compound **2** as an *N*-AcCys adduct of **3**, the absolute configuration of the pyranonaphtoquinone core of **2** was judged to be the same as that reported for **3** (i.e., (+)-griseusin A). For the same reason, the absolute configuration of the α-forosamine moiety was assumed to be D, as found in **3**. This assumption was further supported by the presence in the BGC 1.31 of a set of genes responsible for the biosynthesis of d-forosamine, as discussed below. Bearing in mind the origin of the *N*-AcCys moiety in mycothiol [6,7], we assumed a *(R)* configuration at C-2″′ (i.e., *N*-Ac-l-cysteine). Finally, considering the stereospecific lactonization from the β-face of the pyran C ring generally observed for *γ*-lactone-pyranonaphtoquinones (and more specifically in **3**), we concluded the (*4R*)-configuration in **2** to be reasonable.

Attempting to confirm the origin of **2** (beyond the oxidation from **1**), the LC-HRESIMS chromatogram of the culture extract of *Streptomyces* sp. CA-256286 (pRM4-SARPs) in MO16 medium was interrogated for the presence of the putative parent compound **3**. Interestingly, a [M + H]^+^ ion at *m*/*z* 586.2277 indicative of the molecular formula C_30_H_35_NO_11_ (Δ −1.0 ppm) was indeed detected (Appendix A). Furthermore, the HRMS/MS spectrum revealed the same fragment ion corresponding to the forosamine moiety that was found in **2** (Appendix A). Overall, we reasonably concluded that compound **2** (4-AcCys-FGA) originates from **3** (FGA) as a result of mycothiol addition and further processing, which ultimately confirmed the above assumptions.

Similar AcCys adducts of kalafungin and dihydrokalafungin have been previously reported [20,21] and its occurrence linked to the recruitment of the mycothiol-dependent detoxification pathway found in actinobacteria. Mycothiol (MSH) is the major thiol compound present in certain Gram-positive bacteria, including streptomycetes, and it is used to protect the cells against toxic or reactive electrophiles, thus having analogous functions to glutathione [28]. MSH (its free thiol group) can react with different toxic compounds and form MSH-conjugates, which are then cleaved by a specific amidase to release GlcN-Ins (1-d-myo-inosytyl 2-amido-2-deoxy-α-d-glucopyranoside) and a toxin AcCys S-adduct [29,30].

Based on these statements, the formation of AcCys adducts **1** and **2** from **3** can be reasoned as follows: first, addition of mycothiol to the parent compound **3** and further amidase-mediated cleavage would result in the AcCys S-conjugate **1** (Figure 3). Then, the keto-enol tautomerism of **1** would provide the driving force required for the stereospecific lactonization onto C-4 (Figure 3, I). The intermediate hydroquinone form resulting from this lactonization (not shown) is unstable and would be readily oxidized to the final quinone compound **2**.

The identification of 4-AcCys adducts **1** and **2** may imply that FGA (**3**) is somewhat toxic to the producing strain and represents an example of the export of unwanted metabolites through the action of the mycothiol-dependent detoxification pathway.

When griseusin A and 3′-*O*-α-d-forosaminyl-(+)-griseusin A (**3**) were first discovered, their antimicrobial activity was tested against a panel of different pathogens. It was found that compound **3** had activity against Gram-positive pathogens, with minimal inhibitory concentrations (MICs) values of 0.39 µg/mL for *Staphylococcus aureus* Smith, 0.78 µg/mL for methicillin-resistant *Staphylococcus aureus* No. 5 and 0.39 µg/mL for *Bacillus subtilis* PCI 219 [19]. Recently, the chemical synthesis of griseusin A has been determined and cytotoxicity of griseusin compounds against cancer cells and in axolotl embryo tail inhibition studies has been elucidated, with promising results [25].

On the contrary, compound **2** showed no activity against MSSA ATCC29213 or MRSA MB5393 at the highest concentration tested of 128 µg/mL (Appendix A). This lack of antibacterial activity supports the hypothesis that the AcCys adduct **2** is a detoxified version of 3′-*O*-α-d-forosaminyl-(+)-griseusin A (**3**).

### 2.3. Sequencing of CA-256286, Genome Mining and Identification of Candidate BGCs

The genome of strain CA-256286 was de novo sequenced and is available in NCBI GenBank with accession numbers CP071044 (chromosome) and CP071045 (plasmid) [31]. The NCBI locus tags for the gene names used in the analyses are listed in Appendix A. Taxonomic placement with GTDB-tk [32] identified CA-256286 as a member of *Streptomyces microflavus*, with the representative strain *S. microflavus* JCM 4496 (GCF_014650075.1) having an average nucleotide identity (ANI) of 97.63%. The database strain with the highest ANI is *Streptomyces* sp. NRRL S-623 (99.4%), determined by AutoMLST [33]. Thus, we can confidently say that strain CA-256286 belongs to the genus of *Streptomyces* and from here onwards is referred to as *Streptomyces* sp. CA-256286. Genome mining analysis of CA-256286 with antiSMASH version 5.1.0-ba887d4, with default parameters, predicted that the two contigs have 38 and 3 regions, respectively [34,35]. These regions cover 6 PKS, 5 NRPS, 7 PKS/NRPS, 8 terpene, 1 melanin, 5 bacteriocin/lassopeptide, 2 butyrolactone, 2 siderophore, 2 lanthipeptide, and 3 ectoine BGCs (Appendix A).

Griseusin was first isolated from *Streptomyces griseus* and described back in 1976 [26] and A type II PKS, putatively responsible for griseusin production, was partially sequenced in the genome of *S. griseus* K-63 in 1994 [19,36]. Five genes (ketosynthase, acyl carrier protein, ketoreductase, cyclase, and dehydrase) are available online (NCBI GenBank X77865.1 with the minimal MIBiG entry [37]: BGC0000231). Therefore, we searched for homologs of these genes in the genome of *Streptomyces* sp. CA-256285 and found two candidate type II PKS BGCs that show significant similarity. BGC 1.20 show approximately 72.4–82.6%, and BGC 1.31 show 68.8–79.2% similarity on DNA level to the individual genes of the original griseusin BGC (Table 3).

The “known cluster BLAST” analysis integrated in antiSMASH indicated that BGC 1.20 is only weakly similar to other entries in the MIBiG dataset, with the following top three hits: prejadomycin/rabelomycin/gaudimycin C/gaudimycin D/UWM6/gaudimycin A (27% of genes show similarity); BGC0000201, auricin (44% of genes show similarity); and BGC0000253, oviedomycin (50% of genes show similarity). The same is the case for BGC 1.31, with the following top three hits: BGC0001960, hiroshidine (41% of genes show similarity); BGC0000212, cinerubin B (31% of genes show similarity); and BGC0001074, cosmomycin D (32% of genes show similarity).

### 2.4. Cultivation and Sampling for Omics Studies

In order to determine which of the two identified candidate BGCs is responsible for the production of griseusins, we decided to carry out proteomics and transcriptomics studies. The strains *Streptomyces* sp. CA-256286 with pRM4-SARPs and *Streptomyces* sp. CA-256286 with “empty” pRM4 were cultivated to confirm production and for sampling cells for proteomics and transcriptomics analysis. The cultivations were carried out with five technical replicates, in 50 mL liquid MO16 medium for 7 days at 30 °C with 200 rpm shaking, to imitate the initial cultivation conditions. OD_600_ measurements were taken every two hours for the duration of 18 h and samples for proteomics and transcriptomics were collected at the time point of 10 h, during the exponential growth phase (Figure 4).

Differential production of compound (**1**) with a positive ion *m*/*z* 749 when comparing *Streptomyces* sp. CA-256286 (pRM4-SARPs) and *Streptomyces* sp. CA-256286 (pRM4) was confirmed by extraction in acetone and DMSO, and HPLC-HRESIMS analysis (Figure 5).

### 2.5. Proteomics and Transcriptomics Analyses Confirms Identity of the Griseusin BGC

After identifying and characterizing the produced compound with MS and NMR and proposing two candidate BGCs using genomics, we investigated if the true responsible BGC can be identified using proteomics and transcriptomics data. In the proteomics analysis, whole protein is isolated and digested into small peptides, which are subsequently analyzed via MS-based peptide sequencing and mapped to the genome. In the transcriptomics analysis the transcripts or messenger RNAs (mRNAs) are analyzed. Evaluating the level of transcription and translation of genes in candidate BGCs between a producing and a non-producing strain, often indicates which BGC is expressed [38].

Thus, to determine if expression of BGCs 1.20 or 1.31 correlates with griseusin-production, samples from the 10 h time point in the cultivation (Figure 4) of *Streptomyces* sp. CA-256286 (pRM4-SARPs) and *Streptomyces* sp. CA-256286 (pRM4), were harvested, extracted, and subjected to proteomics analysis. See Appendix A for sample information. By comparing the peptide abundances of the PKS chain length factor (CLF) and ketosynthase (KS) proteins from BGC 1.20 (locus tags FBHECJPB_03027 and FBHECJPB_03028, respectively) and from BGC 1.31 (locus tags FBHECJPB_06071 and FBHECJPB_06072, respectively) it became evident that only the CLF and KS of BGC 1.31 were highly abundant in the strain carrying pRM4-SARPs. No peptides from the core PKS genes in BGC 1.20 were detected and we thus expect that no expression is happening from this cluster. The relative abundance of the CLF peptides from BGC 1.31 is increased 11.95-fold, and in the case of KS peptides from BGC 1.31 it is increased 35.68-fold in *Streptomyces* sp. CA-256286 with pRM4-SAPRs compared to the control strain *Streptomyces* sp. CA-256286 with pRM4. This data clearly shows that the peptides are significantly enhanced and the proteomics data, thus, suggests that the expression of BGC 1.31 is activated by overexpression of SARP family regulators.

To confirm the suggestion that BGC 1.31 is activated and is responsible for the produced griseusins, we also carried out transcriptomics analysis. RNA was purified from the time point of 10 h and sequenced (Novogene, Cambridgeshire, UK). After clean-up of the raw transcriptomics data, differential expression between *Streptomyces* sp. CA-256286 (pRM4-SARPs) and *Streptomyces* sp. CA-256286 (pRM4) was analyzed using ReadXplorer [39,40] and CLC genomics (QIAGEN, version 12.0.3). We compared the differential expression of all genes from BGC 1.20 and 1.31 and illustrated the data using heat maps (Appendix A). For BGC 1.20 the heat map does not show any obvious patterns and the expression of the core PKS genes remain unchanged in both the strains expressing the SARP regulators and the controls. For BGC 1.31, most of the genes are clearly expressed in *Streptomyces* sp. CA-256286 (pRM4-SARPs) and not in *Streptomyces* sp. CA-256286 (pRM4), including the two core type II PKS genes with locus tags FBHECJPB_06071 and FBHECJPB_06072. A combined heat map of the proteomics and transcriptomics data for all genes in the predicted BGC 1.31 was generated (Figure 6), which clearly shows an upregulation of the majority of the genes in the strain with overexpressed SARP family regulators. Based on the transcriptomics and proteomics analysis, we thus had good indications that BGC 1.31 is responsible for the production of compound (**3**) 3′-*O*-α-d-forosaminyl-(+)-griseusin A.

### 2.6. Gene Inactivation of the Griseusin PKS KS/CLF by CRISPR-cBEST Base Editing

To experimentally confirm that BGC 1.31 is responsible for the griseusin production, the two core type II PKS genes with locus tags FBHECJPB_06072 (KS) and FBHECJPB_06071 (CLF) were inactivated using CRISPR-cBEST base editing [41,42]. The introduced base changes, resulting in stop-codons, were confirmed by PCR and sequencing (Figure 7). The strains were cured of the temperature sensitive CRISPR-cBEST plasmids by cultivation at 37 °C and re-streaking on medium without apramycin selection. Both mutant strains did not produce the griseusin compounds, neither with nor without overexpression of the SARP family regulators (Figure 8) indicating the involvement of the KS and CLF genes from BGC 1.31 in the biosynthesis.

In order to verify that this effect is truly due to the gene inactivations and not potential CRISPR off-target effects, we carried out complementation experiments. The KS-encoding gene FBHECJPB_06072 and the CLF-encoding gene FBHECJPB_06071 were cloned separately on the integrative plasmid pRM4. For expressing the SARP family regulators along with other plasmids with apramycin resistance, e.g., these complementations, the SARP genes *actIIORF4-griR-aur1PR3-papR2-redD* were subcloned on the replicative plasmid pKC1218 [43] (Prof. S. Zotchev, Uni Vienna) carrying a hygromycin resistance cassette. We verified that expression of the SARP regulators from both pRM4-SARPs and pKC1218-SARPs was sufficient to activate expression of the two compounds (**3**) 3′-*O*-α-d-forosaminyl-(+)-griseusin A and the AcCys adduct **1** (Appendix A).

The plasmids encoding the complementation of CLF and KS were transferred to the edited strains cured of CRISPR-cBEST plasmids along with pKC1218-SARPs. Cultivation in MO16 medium showed that this complementation was enough to re-active production of compound **1** (Figure 8). There was no production either in the inactivated stains with only the SARP regulators or with only the complemented gene, thus the production is solely activated with a functional copy of both genes and the SARP regulators present. This observation experimentally confirms that BGC 1.31 is responsible for the production of 3′-*O*-α-d-forosaminyl-(+)-griseusin A (**3**).

### 2.7. Heterologous Expression of the Putative Griseusin BGC in the Heterologous Host S. albus J1074

A BAC library based on pESAC-13-Apramycin [44] was constructed and a clone containing the complete BGC 1.31 was identified (Bio S&T Inc., Saint-Laurent, QC, Canada). This BAC was transferred to Streptomyces albus J1074 via three-parental conjugation [45]. Cultivation of this heterologous host carrying the BAC in MO16 medium showed no production of the griseusin compounds. Expressing both the BAC and the SARP transcriptional regulators on pKC1218-SARPs resulted in production of griseusins in the heterologous host (Figure 9). This further supports that cluster 1.31 is encoding all responsible genes for the production of 3′-*O*-α-d-forosaminyl-(+)-griseusin A (3). Interestingly, we also observe production of the modified compound 1 when heterologously expressing BGC 1.31. If the addition of mycothiol is indeed a detoxification mechanism, this suggests that either the genes for the detoxification pathway are encoded in BGC 1.31 or that they are natively present in the genome of *S. albus* J1074. This is further discussed in the section on biosynthetic pathway prediction in Section 2.9.

### 2.8. Identification of the Responsible Activator

Our results indicate that one or more of the SARP family regulators that are encoded in pRM4-SARPs (*actIIORF4*, *griR*, *aur1PR3*, *papR2* or *redD)* is responsible for the transcriptional activation of BGC 1.31. Comparing the protein sequences of the cloned SARPs to two SARPs predicted in BGC 1.31 using BLASTP (NCBI), all five SARP genes show similarity to both FBHECJPB_06080 (query covers 83–95% and perc. ident. 30–41%) and FBHECJPB_06090 (query covers 86–91% and perc. ident. 32–46%). None of the five cloned SARP genes are thus significantly more similar to one of the two SARPs in the BGC, and we thus decided to test all five experimentally. Plasmids individually encoding one of the five SARP family transcriptional regulators were transferred to CA-256286 and the strains were cultivated in MO16 medium. This experiment demonstrated that only ActII-ORF4 could activate production of compound (**1**) and therefore is responsible for activating griseusin production in *Streptomyces* sp. CA-256286 (Appendix A).

### 2.9. The Griseusin Biosynthesis Pathway

After confirming that BGC 1.31 is responsible for the production of 3′-*O*-α-d-forosaminyl-(+)-griseusin A (**3**), which is subsequently modified into the AcCys adduct compound **1**, we aimed to reconstruct the putative biosynthetic pathway based on the collected structural, -omics and bioinformatic information.

First, we looked into the formation of AcCys adducts **1** and **2** from 3′-*O*-α-d-forosaminyl-(+)-griseusin A (**3**). A mycothiol-dependent detoxification pathway is characterized in *S. coelicolor* A3(2) [20], covering five genes; *mshA* (SCO4204), *mshB* (SCO5126), *mshC* (SCO1663), and *mshD* (SCO4151), along with Mca (SCO4967) a MSH S-conjugate amidase [46,47]. We hypothesized that these genes might be similar to genes needed for the modification of **3** to **1** and **2**, and we thus searched for homologs in the genome of CA-256286 (Table 4). The identified homologs show 70.4–88.3% similarity and are not clustered together or located close to BGC 1.31.

Since we observed both, the parent compound (**3**) 3’-*O*-α-d-forosaminyl-griseusin A and the AcCys adduct **1**, when heterologously expressing BGC 1.31 in host *S. albus* J1074, we analyzed if there were also similar genes in the genome of *S. albus* J1074 (Appendix A). Indeed, homologs with protein identities of 71.7 to 87.2% were also identified in this strain. The detoxification genes thus seem to be conserved between different species. We hypothesize that when the parent compound is produced above a certain level, the detoxification is necessary as a self-immunity mechanism. The detoxification genes are, however, not necessarily part of the BGC for the metabolites being detoxified.

The core carbon structures of aromatic polyketides are synthetized by a minimal PKS complex of a ketosynthase (KS) and chain length factor (CLF) dimer (KS-CLF) catalyzing the condensation reactions, while the chain is tethered to an acyl carrier protein (ACP) [48,49]. Several known pyranonaphthoquinone polyketides are described and their biosynthesis is studied [16]. Pyranonaphthoquinones are composed of three rings; a pyran, a quinone, and a benzene, as for the griseusins. Determining the genes from BGC 1.31 involved in the biosynthesis of the parent compound (**3**) 3’-*O*-α-d-forosaminyl-griseusin A, we started by looking at the details for the predicted BGC in antiSMASH. Additionally, from the minimal/core PKS genes already confirmed (locus tags FBHECJPB_06071 and FBHECJPB_06072) and the ACP (FBHECJPB_06070), we would expect to find genes for specific enzymes based on the chemical structure (Appendix A). Two cyclases are likely needed for the formation of the two 6 membered rings, where a C7-C12 (FBHECJPB_06069) and a C5-C14 (FBHECJPB_06079) cyclase were predicted in antiSMASH. These fit perfectly with the chemical structure of ring A and B (Figure 1). For actinorhodin, it was found that the C7-C12 cyclization could happen in the active site of the KS-CLF complex and could be followed by reduction of the ketone on C-9, aromatization and cyclization of the second ring before the bicyclic intermediate is released [50]. Taking this information together with the predicted cyclases in antiSMASH, we believe that the cyclizations in (**3**) happen in the order C7-C12 and then C5-C14. From the compound structure there should be an O-methyltransferase (O-MT), but none are predicted, only two methyltransferases (MTs); FBHECJPB_06067/desVI (similar to MT from spiramycin BGC) and FBHECJPB_06098. InterProScan results for both indicate that they belong to a S-adenosyl-l-methionine-dependent methyltransferase superfamily. Ketoreductase (KR) and dehydratase (DH) enzymes are needed to reduce some of the keto-groups, and possibly also in cyclization reactions. KRs are predicted in the genes FBHECJPB_06064, FBHECJPB_06084, FBHECJPB_06086 and FBHECJPB_06088. See Appendix A for an overview of all genes predicted in BGC 1.31 and the proposed functions.

Compound **3** (3′-*O*-α-d-forosaminyl-(+)-griseusin A) has a forosamine sugar attached at C-3´ within the ring E. The biosynthesis of deoxysugars related to forosamine and the transfer to pyranonaphthoquinones are common in aromatic polyketides [16]. This transfer is most often dependent on a glycosyltransferase (GT), for which there is only one predicted (FBHECJPB_06065) in BGC 1.31, and we thus believe that this is the GT responsible for attaching the 3′-*O*-α-d-forosaminyl part. Biosynthesis of TDP-d-forosamine in the spinosyn pathway has been characterized [51] and is carried out by the enzymes SpnO, SpnN, SpnQ, SpnR and SpnS. To determine if homolog genes are present in BGC 1.31, we used BLAST analysis against the genome of *Streptomyces* sp. CA-256286 and found homolog genes for all 5 enzymes in BGC 1.31 (Table 5). This gives further hints to which genes are involved in the biosynthetic pathway of 3′-*O*-α-d-forosaminyl-(+)-griseusin A (**3**).

The polyketide auricin, produced by *S. aureofaciens* CCM 3239, is structurally closely related to the griseusins [52]. The auricin BGC *aur1* is located on a large linear plasmid (MIBiG accession BGC0000201). Interestingly, the transcriptional regulation of the auricin BGC *aur1* is very complex and controlled by several different regulators [52,53]. One SARP family regulator from *aur1* (*aur1PR3*) is actually included on the SAPR overexpression plasmids, but this did not activate expression of BGC 1.31. Like the griseusins, auricin also has a d-forosamine sugar attached. The genes involved in the biosynthesis of the forosamine moiety and transfer were described, and it was found that two GTs are responsible for the transfer [54].

A BGC alignment of the auricin BGC (MIBiG entry BGC0000201), the fragment of the originally described *S. griseus* griseusin BGC (MIBiG BGC0000231), and BGC 1.31 using clinker [55] (Figure 10), showed that despite being responsible for the biosynthesis of highly similar structures, the *S. griseus* griseusin BGCs and the auricin BGC are surprisingly different to BGC 1.31. To extend this analysis to other strains, we compared BGC 1.31 from CA-256286 against a large dataset of BGCs from 212 complete high-quality *Streptomyces* genomes and the known BGCs from the MIBiG reference database [56] using BiG-SCAPE [57] (with cutoffs up to 0.5), which did not result in any significant hit. Next, we expanded the search to the BiG-FAM database [58] of gene cluster families, which also includes draft genomes. The BGC 1.31 was found to be a member of the GCF_25160 family, which additionally contained two BGCs from draft genomes of *Streptomyces* sp. NRRL S-623 and *Streptomyces* sp. 2R. Aligning the three clusters proves that they are almost completely identical (Figure 10) to the BGC from CA-256286. The entire genome of CA-256286 is 99% similar to that of *Streptomyces* sp. NRRL S-623 and it is probably the same species.

## 3. Conclusions

In this study, we have demonstrated the effectiveness of applying transcriptional regulator-carrying plasmids for induction of cryptic metabolites in streptomycetes. The established workflow can be easily adapted for use in other actinobacteria, with the only requirement that they are genetically accessible. As a proof of principle, we have shown production of a novel *N*-acetyl cysteine adduct of 3′-*O*-α-d-forosaminyl-(+)-griseusin A, a previously described polyketide antibiotic. As hypothesized, this compound was not produced by the wild type, but could be induced by introduction of plasmid-encoding *Streptomyces* antibiotic regulatory protein (SARP) family regulators. Using a combination of multi-omics approaches and heterologous expression, we were able to identify a type II PKS BGC coding for the production of this metabolite. Interestingly, the identified BGC differs significantly from a previously studied auricin BGC. Biosynthesis mechanisms of griseusin and its modified derivatives remain unclear. We plan to further investigate these by using genetic engineering techniques for manipulation of the wild type producer, which were established in this study.

## 4. Materials and Methods

### 4.1. Strains and Cultivations

All *Escherichia coli* transformations were carried out according to manufacturer’s instructions. Commercially competent *E. coli* One Shot^®^ Mach1™ (Thermo Fisher Scientific, Carlsbad, CA, USA), DH10beta or DH5alpha were used for cloning purposes. *E. coli* were grown on solid Lysogeny broth (LB) plates or liquid LB medium supplemented with appropriate antibiotics at 37 °C. Non-methylating *E. coli* ET12567 with pUZ8002 were used for transfer of plasmids by conjugation into *Streptomyces* strains [12]. A large panel of bioactive actinomycetes strains was received from Fundación MEDINA (Spain). *Streptomyces albus* J1074 (Prof S. Zotchev (Uni Vienna)) was used as a heterologous host. Three parental conjugations were carried out for transfer for large BACs, where *E. coli* BW 25113 with pUB307 mobilization plasmid (Xinglin Jiang) was used. Soy flour mannitol plates supplemented with 10 mM MgCl_2_ and appropriate antibiotics were used for conjugations and routine cultivations. ISP2 liquid medium was used for pre-cultures. All *Streptomyces* cultivations were carried out at 30 °C. MO16 medium were used for cultivations (Glucose 10 g/L, Soluble starch from potato 10 g/L, Maltose 10 g/L, Bacto yeast extract 1 g/L, Bacto soytone 5 g/L, Bacto tryptone 4 g/L, KH_2_PO_4_ 0.1 g/L, K_2_HPO_4_ 0.2 g/L, MgSO_4_ · 7H_2_O 0.05 g/L, NaCl 0.02 g/L, CaCl_2_ · 2H_2_O 0.05 g/L, Trace Mix M003 1 mL/L, pH adjusted to 5.8). Trace Mix M003 (SnCL_2_ · 2H_2_O 0.005 g/L, H_3_BO_3_ 0.01 g/L, Na_2_MoO_4_ · 2H_2_O 0.012 g/L, CuSO_4_ 0.015 g/L, CoCl_2_ · 6H_2_O 0.02 g/L, KCl 0.02 g/L, ZnCl_2_ 0.02 g/L, MnSO_4_ · 4H_2_O 0.1 g/L, FeCl_3_ 5.8 g/L, HCl 2 mL/L). The integration site might exist more than once in the genome, as pseudo-sites [59], but only one transconjugant was studied from each conjugation. Samples for transcriptomics and proteomics analyses were collected from MO16-grown cultures of strains to be compared. Strains were grown in 50 mL baffled shake flasks in pentaplicate at 30 °C and 180 rpm shaking speed. Only the samples collected in the exponential growth phase were analyzed in this study. The exponential growth phase was determined from the OD600-based growth curve built for both strains (Figure 4). For RNA isolation 1 mL of culture was collected from each replicate, followed by 5 s centrifugation at top speed, discarding of supernatant, and immediate freezing of the cell pellet in liquid nitrogen. For proteomics, 5 mL culture was centrifugated and the cell pellet frozen at −20 °C. Samples for transcriptomic and proteomic analyses were collected simultaneously from corresponding cultures to allow complementary analysis of both datasets.

### 4.2. General Molecular Biology Techniques

All oligos were purchased from Integrated DNA Technologies (IDT) (Leuven, Belgium). PCR reactions were carried out with Q5 polymerase (New England Biolabs, Ipswich, MA, USA). Fragments were analyzed on 1%TAE-agarose gels. Sanger sequencing was performed with Eurofins Genomics Mix2Seq kits.

### 4.3. Plasmids

The SARP family regulators (pRM4-ermE-actIIORF4-griR-aur1PR3-papR2-redD) were cloned into plasmid pKC1218 (Prof S. Zotchev (Uni Vienna)) using Gibson assembly.

### 4.4. BAC Library and Heterologous Expression

A BAC library based on pESAC-13-apramycin was constructed by Bio S&T Inc. (Saint-Laurent, QC, Canada), and a clone containing BGC 1.31 was screened and identified by them based on PCR. The identified BAC was transferred to *S. albus* J1074 according to a standard conjugation protocol [12].

### 4.5. Comparative Metabolomics with LC-MS

Cultures were extracted in 1:1 acetone, shaken for 2 h at 200 rpm and centrifuged to remove cell debris; then 0.03 mL DMSO per 1 mL extraction was added. The extracts were evaporated to approximately ⅓ of the initial cultivation volume with a gentle nitrogen stream or using a rotary evaporator.

Comparative metabolite profiling was performed using an Agilent (Agilent Technologies, Santa Clara, CA, USA) 1290 Infinity ultrahigh-performance liquid chromatography (UHPLC) system coupled to an Agilent UV/Vis diode array detector (DAD; 190.0–640.0 nm) and an Agilent 6545 quadrupole time-of-flight (Q-TOF) mass spectrometer using electrospray ionisation (ESI). The LC stationary phase used an Agilent Poroshell 120 Phenyl-Hexyl (2.1 × 150 mm, 1.9 micron). Separation was achieved with a water-acetonitrile (ACN) gradient mobile phase (gradient: 0.0–10.0 min, 10% to 100% B; isocratic: 10.0–12.0 min, 100% B; gradient: 12.0–12.1 min, 100% to 10% B; isocratic: 12.1–14.0 min, 10% B), at 0.350 mL/min flow rate and the temperature set at 40 °C. MS data were recorded in positive ionization (+ESI mode) with a mass range (*m*/*z*) of 100–1700, and a scan rate of 10 spectra/s. MS/MS fragmentation was achieved using data-independent acquisition (DIA) with fixed collision energies at 10, 20, and 40 CeV, with precursor ions selected for fragmentation based on abundances with a threshold of 5000 counts (Abs). Data analysis was performed using MassHunter (Agilent Technologies; v.B06.00).

The instrumentation used for the analysis of the samples in Figure 4 is a Dionex Ultimate 3000 ultra-high-performance liquid chromatography (UHPLC) coupled to a UV/Vis diode array detector (DAD) in the range 200–700 nm and a high-resolution mass spectrometer (HRMS) Orbitrap Fusion (ThermoFisher Scientific, Waltham, MA, USA). The UHPLC method used for the analysis was the following: column, Zorbax Eclipse Plus C-18 column (2.1 × 100 mm, 1.8 μm) (Agilent, Santa Clara, CA, USA); column temperature: 35 °C; solvent A (H_2_O buffered with 0.1% HCOOH) and solvent B (CH_3_OH buffered with 0.1% HCOOH); isocratic: 0–0.6 min, 5% B; gradient: 0.6–13 min, 5–95% B; isocratic: 13–15.5 min, 95% B; gradient: 15.5–15.6 min, 95–5% B; isocratic: 15.6–17.5 min, 5% B; and flow rate, 0.350 mL/min. The HRMS was performed in positive mode (+ESI), at 3500 V spray voltage, in the mass range (*m*/*z*) 100–1000 at a resolution of 120K, RF Lens 50%, and AGC target 200K. Before analysis, the MS was calibrated using ESI Positive ion Calibration Solution Pierce™ LTQ Velos ESI Positive Ion Calibration Solution. The software Xcalibur 4.2 (Thermo Fisher Scientific Inc., Waltham, MA, USA) was used for data analysis.

### 4.6. General Experimental Procedures

Solvents employed in this work were all HPLC grade. Optical rotations were measured on a Jasco P-2000 polarimeter. IR spectra were recorded with a JASCO FT/IR-4100 spectrometer equipped with a PIKE MIRacle single reflection ATR accessory. LC-UV-LRMS analyses were performed on an Agilent 1100 single quadrupole LC-MS system as previously described [60]. HRESIMS and MS/MS spectra were acquired using a Bruker maXis QTOF mass spectrometer coupled to an Agilent Rapid Resolution 1200 LC. The mass spectrometer was operated in positive ESI mode. The instrumental parameters were 4 kV capillary voltage, drying gas flow of 11 L min^−1^ at 200 °C, and nebulizer pressure of 2.8 bar. TFA-Na cluster ions were used for mass calibration of the instrument prior to sample injection. Pre-run calibration was done by infusion with the same TFA-Na calibrant. Medium pressure liquid chromatography (MPLC) was performed on semiautomatic flash chromatography (CombiFlash Teledyne ISCO Rf400×) with a precast reversed-phase column. Semipreparative HPLC separation was performed on a Gilson GX-281 322H2 chromatographic system. NMR spectra were recorded on a Bruker Avance III spectrometer (500 and 125 MHz for ^1^H and ^13^C NMR, respectively) equipped with a 1.7 mm TCI MicroCryoProbe. Chemical shifts were reported in ppm using the signals of the residual solvents as internal reference (d_H_ 2.51 and d_C_ 39.5 for DMSO-d_6_).

### 4.7. Isolation and Characterization Data of N-AcCys Adduct 2

Isolation of AcCys adducts **1** and **2** from a large-scale fermentation: A 2 L culture (16 × 500 mL flasks containing 125 mL of M016 medium) of C-300354 (30 °C, 7 days) was extracted by addition of acetone (2 L), shaken at 220 rpm for 1 h, centrifuged at 9500 rpm, and filtered to discard mycelial debris. The broth extract was concentrated under a nitrogen stream until initial volume (2 L). The aqueous residue was divided in two equal portions which were separately loaded onto two SP207ss columns (65 g, 32 × 100 mm) and eluted with a step gradient of acetone: water (10% acetone for 6 min, 20% acetone for 6 min, 40% acetone for 6 min., 60% acetone for 6 min, 80% acetone for 6 min, then 100% acetone for 10 min; 10 mL/min, 15 mL/fraction). LC-MS analysis identified fractions 12–16 as those containing the target compound. These fractions were combined and further purified by semipreparative HPLC (Waters XBridge Biphenyl, 10 × 150 mm) using a linear gradient of H_2_O: CH_3_CN (both containing 0.1% TFA) from 20% to 30% CH_3_CN-TFA in 35 min (UV detection at 210 and 280 nm), to yield **1** (1.0 mg) as a yellow-orange amorphous powder eluting at 21.5 min. After spontaneous oxidation of **1** to **2**, the latter compound was purified by semipreparative HPLC by using the same method, column, and conditions as described above for **1**. AcCys adduct **2** was thus isolated as an orange amorphous powder (0.7 mg).

3′-*O*-α-d-forosaminyl-(+)-griseusin A, 4-*N*-acetyl-l-cysteine adduct (**2**): [α]^25^_D_ + 57.6 (c 0.18, MeOH); UV (MeOH) λ_max_ 216, 252, 440. IR (ATR) ν_max_ 3396, 2953, 1678, 1441, 1397, 1205, 1136, 1023 cm^−1^; ^1^H and ^13^C NMR data, Table 2; (+)-ESI-TOFMS *m*/*z* 747.2434 [M + H]^+^ (calcd. for C_35_H_43_N_2_O_14_S^+^, 747.2430), 374.1258 [M + 2H]^2+^ (calcd. for C_35_H_44_N_2_O_14_S^2+^, 374.1251).

### 4.8. RNA seq. and Transcriptomics Analysis

The cell pellets were homogenized using NucleoSpin bead tubes type B (Macherey-Nagel), directly followed by RNA isolation using the RNeasy kit from Qiagen. The rRNA depletion, library preparation, and sequencing were carried out by Novogene (Cambridge, UK).

### 4.9. Proteomics

Frozen cells were kept at −80 °C until processing of samples. Thawing of the cells were done on ice and any remaining supernatant was removed after centrifugation at 15,000× *g* for 10 min. While kept on ice two 3-mm zirconium oxide beads (Glen Mills, NJ, USA) were added to the samples. Immediately after moving the samples away from ice 100 μL of 95 °C GuanidiniumHCl (6 M Guanidinium hydrochloride (GuHCl), 5 mM tris(2-carboxyethyl)phosphine (TCEP), 10 mM chloroacetamide (CAA), 100 mM Tris–HCl pH 8.5) was added to the samples. Cells were disrupted in a Mixer Mill (MM 400 Retsch, Haan, Germany) set at 25 Hz for 5 min at room temperature, followed by 10 min in thermo mixer at 95° at 2000. Any remaining cell debris was removed by centrifugation at 15,000× *g* for 10 min, after which 50 μL of supernatant were collected and diluted with 50 μL of 50 mM ammonium bicarbonate. Based on protein concentration measurements (BSA), 100 µg protein were used for tryptic digestion. Tryptic digestion was carried out at constant shaking (400) for 8 h, after which 10 μL of 10% TFA was added and samples were ready for StageTipping using C18 as resin (Empore, 3M, St Paul, MN, USA). For analysis of the samples a CapLC system (Thermo Scientific, Waltham, MA, USA) coupled to an Orbitrap Q-exactive HF-X mass spectrometer (Thermo Scientific) was used. First samples were captured at a flow of 10 μL/min on a precolumn (µ-precolumn C18 PepMap 100, 5 µm, 100Å) and then at a flow of 1.2 µL/min the peptides were separated on a 15 cm C18 easy spray column (PepMap RSLC C18 2 µm, 100Å, 150 µm × 15 cm). The applied gradient going form 4% acetonitrile in water to 76% over a total of 60 min. While spraying the samples into the mass spectrometer the instrument operated in data-dependent mode using the following settings: MS-level scans were performed with Orbitrap resolution set to 60,000; AGC Target 3.0e6; maximum injection time 50 ms; intensity threshold 5.0e3; and dynamic exclusion 25 s. Data-dependent MS2 selection was performed in Top 20 Speed mode with HCD collision energy set to 28% (AGC target 1.0 × 10^4^, maximum injection time 22 ms, isolation window 1.2 *m*/*z*). For analysis of the thermo raw files Proteome discoverer 2.3 was used with the following settings: fixed modifications, carbamidomethyl (C), and variable modifications, oxidation of methionine residues; first search mass tolerance 20 ppm and a MS/MS tolerance of 20 ppm; trypsin as enzyme and allowing one missed cleavage; FDR was set at 0.1%; match-between-runs window was set to 0.7 min; quantification was only based on unique peptides, and normalization between samples was based on total peptide amount.

### 4.10. CRISPR-cBEST Inactivations and Verification

CRISPR-cBEST plasmids were designed and constructed using a standard protocol [41]. The sgRNAs were designed with CRISPy Web tool (crispy.secondarymetabolites.org, 21 February 2020) [42]. The sgRNA sequences were GAACCAGGCGAAGGACTGAT (targeting FBHECJPB_06071) and AACCGACGTGTTGTCATCAC (targeting FBHECJPB_06072). The edits were verified by colony PCR and Sanger sequencing of the PCR product.

### 4.11. Antimicrobial Sensitivity Testing

Compound **2** was tested in antimicrobial assays against the growth of gram-positive bacteria methicillin-resistant *S. aureus* (MRSA) MB5393 and methicillin-susceptible *S. aureus* (MSSA) following previously described methodologies [61]. Vancomycin was used as positive control.

## Figures and Tables

**Figure 1 molecules-26-06580-f001:**
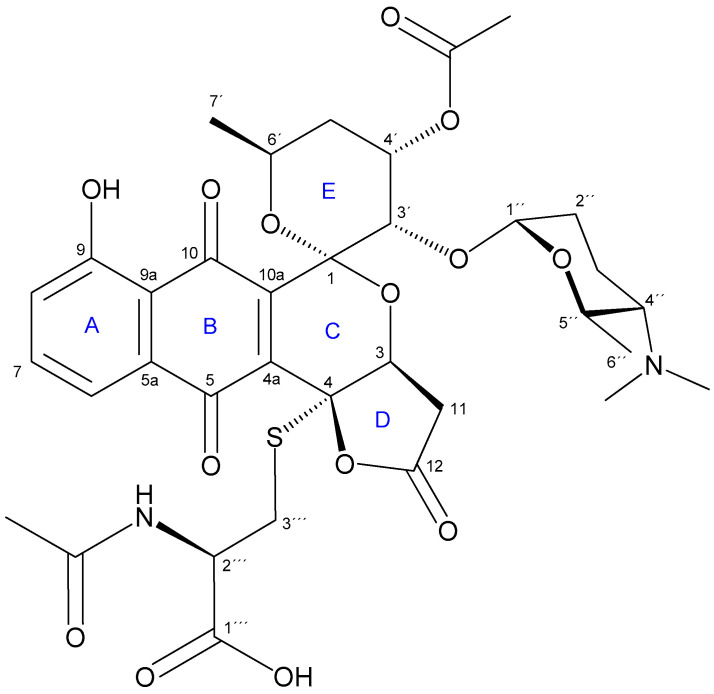
Structure of **2** (4-AcCys-FGA). A–E rings of the *γ*-lactone-pyranonaphtoquinone core are indicated.

**Figure 2 molecules-26-06580-f002:**
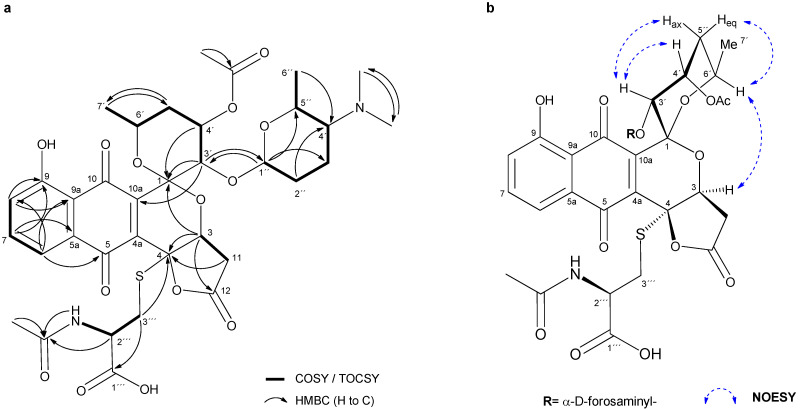
(**a**) Key COSY/TOCSY and HMBC correlations in **2**. (**b**) Key NOESY correlations determining the relative configuration in **2**.

**Figure 3 molecules-26-06580-f003:**
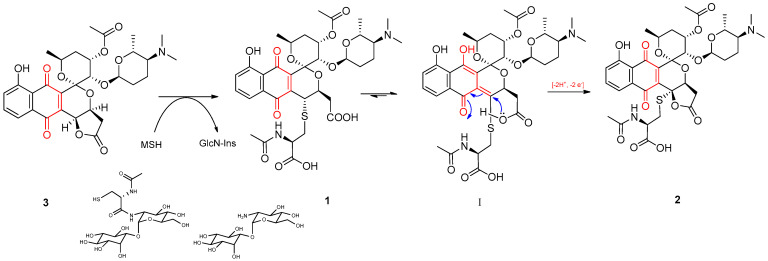
A proposed mechanism for the formation of AcCys adducts **1** and **2** from **3** by recruiting mycothiol.

**Figure 4 molecules-26-06580-f004:**
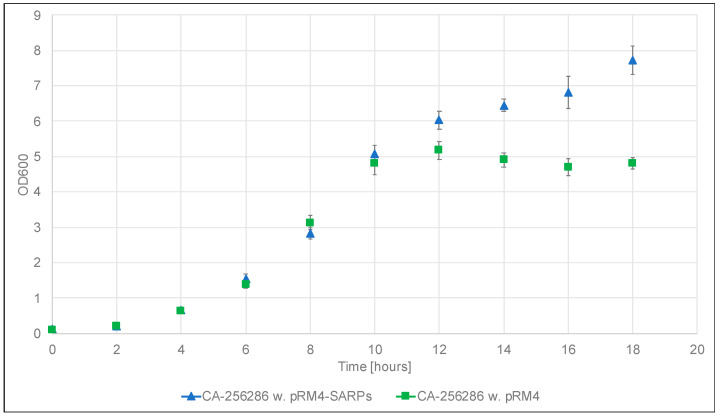
Growth curves of *Streptomyces* sp. CA-256286 with pRM4-SARPs (five replicates, blue triangles) and *Streptomyces* sp. CA-256286 with pRM4 (five replicates, green squares) based on OD_600_ measurements carried out every two hours for 18 h.

**Figure 5 molecules-26-06580-f005:**
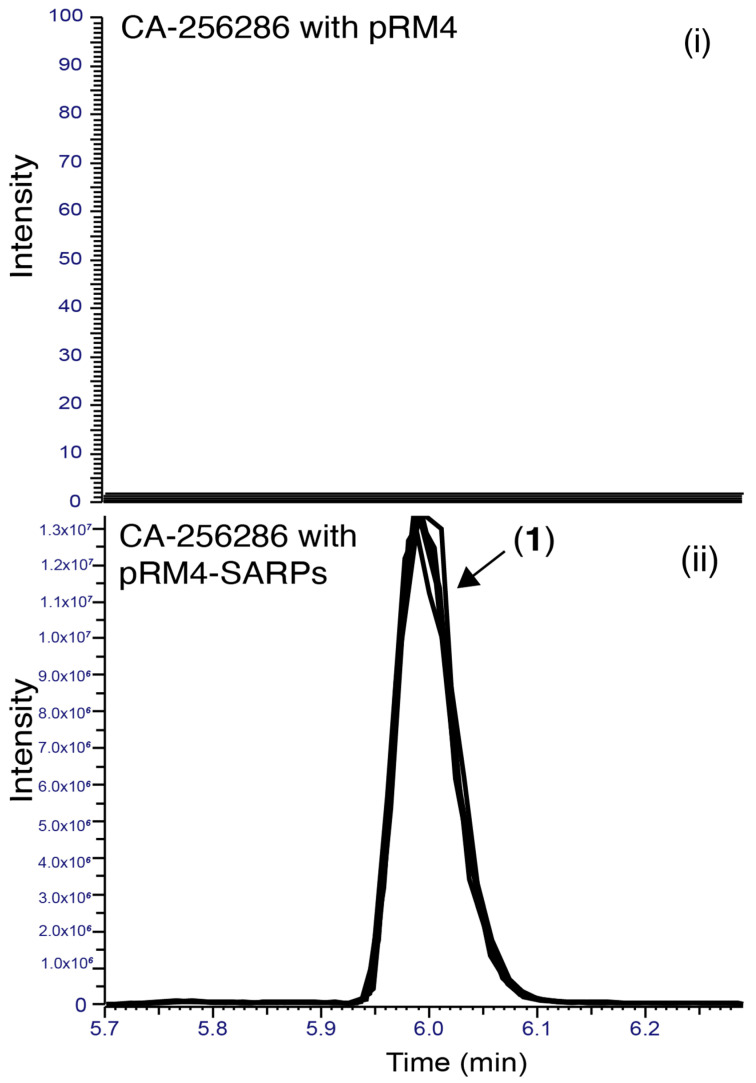
Extracted Ion Chromatograms (EICs) for the detection of compound (**1**) in CA-256286 with pRM4 (**i**) and CA-256286 with pRM4-SARPs (**ii**). Five biological replicates are displayed in overlaid mode.

**Figure 6 molecules-26-06580-f006:**
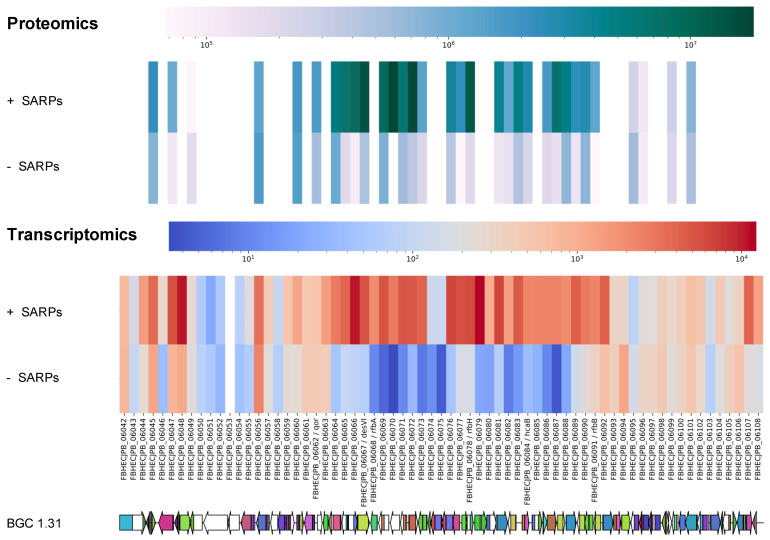
Heat map of peptide and transcript levels of all genes predicted in BGC 1.31 in *Streptomyces* sp. CA-256286 with pRM4-SARPs and *Streptomyces* sp. CA-256286 with pRM4.

**Figure 7 molecules-26-06580-f007:**
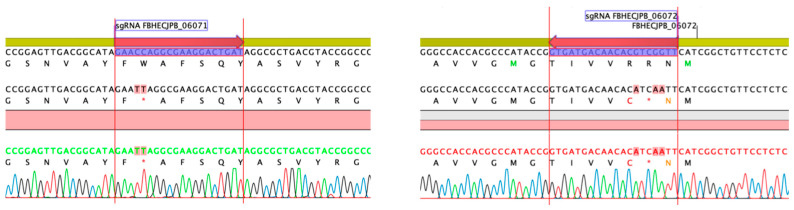
Inactivation of core type II PKS genes FBHECJPB_06071 (CLF) and FBHECJPB_06072 (KS) in BGC 1.31 by introduction of stop codons using CRISPR-cBEST.

**Figure 8 molecules-26-06580-f008:**
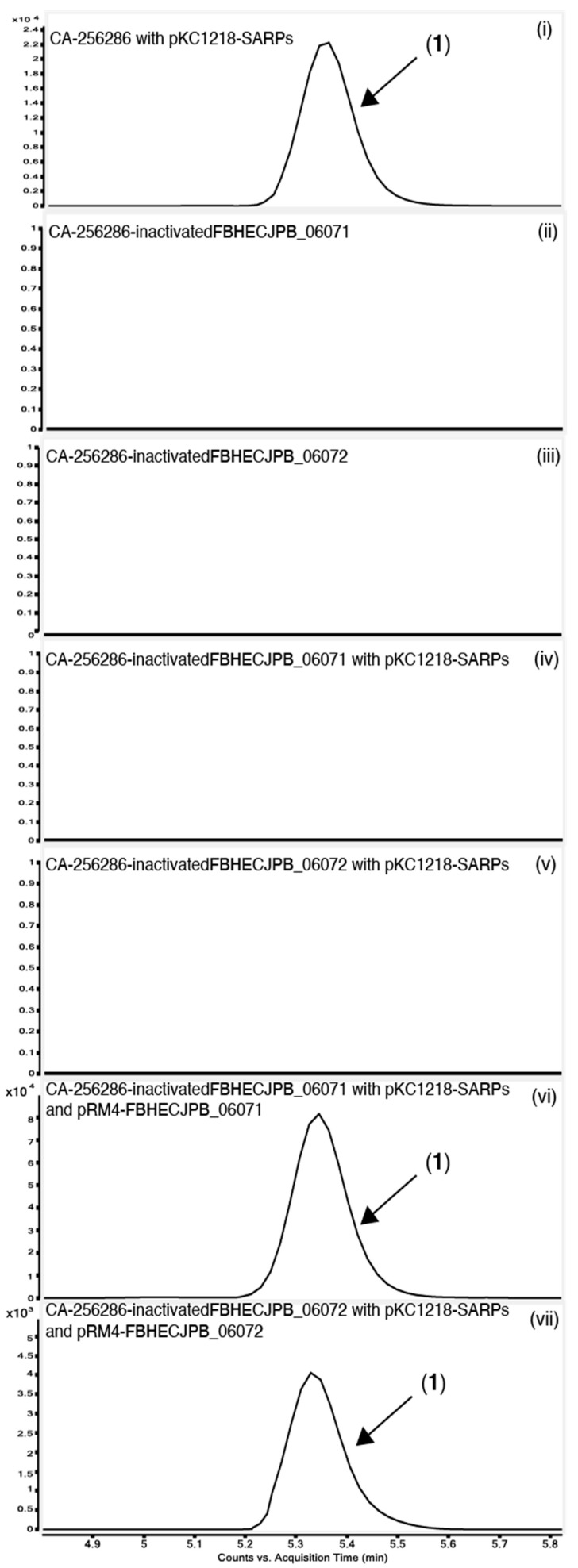
Extracted Ion Chromatograms (EICs) for the detection of compound (**1**) in CA-256286 with pKC1218-SARPs (**i**), CA-256286-inactivatedFBHECJPB_06071 (**ii**), CA-256286-inactivatedFBHECJPB_06072 (**iii**), CA-256286-inactivatedFBHECJPB_06071 with pKC1218-SARPs (**iv**), CA-256286-inactivatedFBHECJPB_06072 with pKC1218-SARPs (**v**), CA-256286-inactivatedFBHECJPB_06071 with pKC1218-SARPs and pRM4-FBHECJPB_06071 (**vi**) and CA-256286-inactivatedFBHECJPB_06072 with pKC1218-SARPs and pRM4-FBHECJPB_06072 (**vii**). One out of three biological replicates is displayed (see Appendix A for further details).

**Figure 9 molecules-26-06580-f009:**
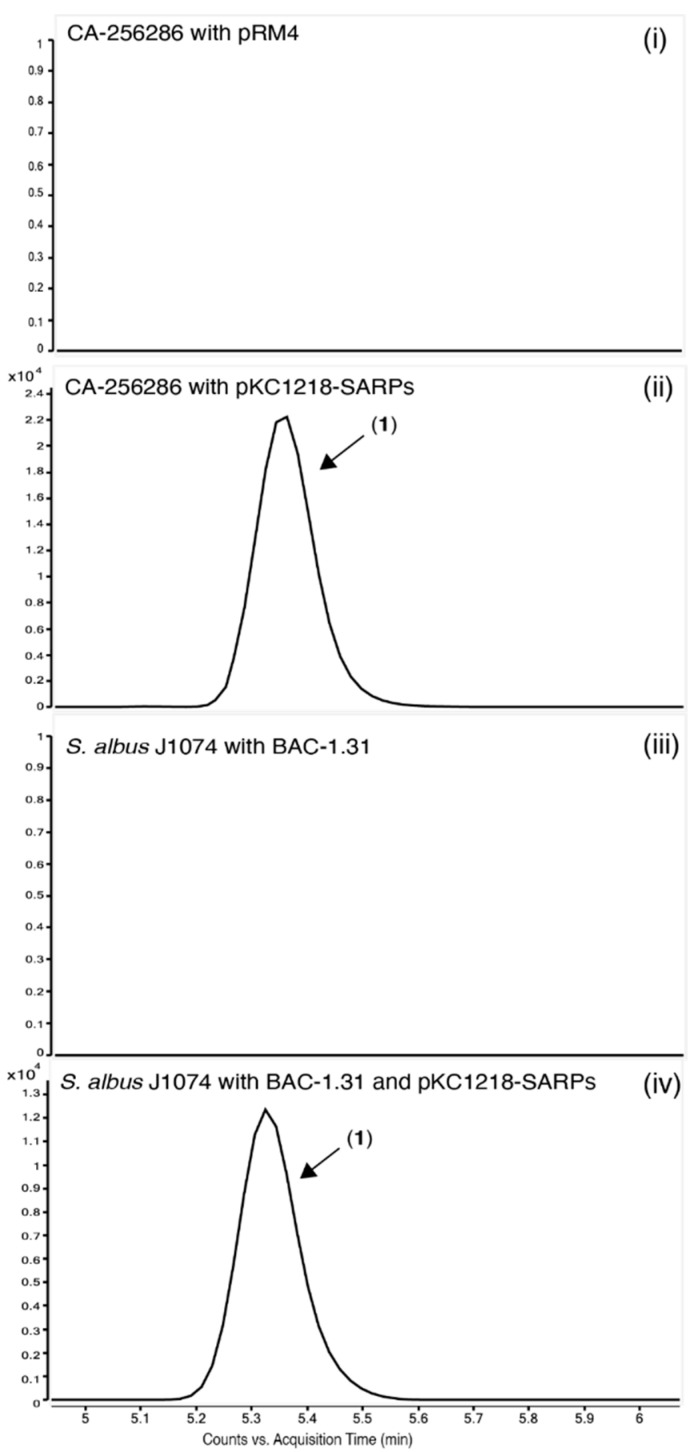
Extracted Ion Chromatograms (EICs) for the detection of compound **1** in CA-256286 with pRM4 (**i**), CA-256286 with pKC1218-SARPs (**ii**), *S. albus* J1074 with BAC-1.31 (**iii**) and *S. albus* J1074 with BAC-1.31 and pKC1218-SARPs (**iv**). One out of three biological replicates is displayed (see Appendix A for further details).

**Figure 10 molecules-26-06580-f010:**
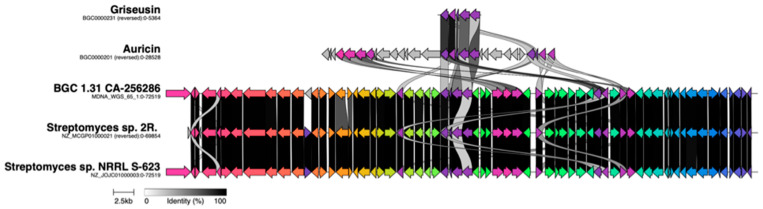
Alignment of BGC 1.31 in CA-256286 against the originally described griseusin BGC from *S. griseus*, an auricin BGC from *S. aureofaciens* CCM 3239, one BGC in *Streptomyces* sp. 2R (NZ_MCGP01000021), and one BGC in *Streptomyces* sp. NRRL S-623 (NZ_JOJC01000003). The dark spaces show similar genes.

**Table 1 molecules-26-06580-t001:** Overview of plasmids encoding transcriptional regulators.

Class of Transcriptional Regulator	Transcriptional Regulator Genes	Plasmid Name
Cluster specific regulators	*aur1P-pntR-strR-fkbN*	pRM4-CSRs/pEM1 [7]
*Streptomyces* antibiotic regulatory proteins	*actIIORF4-griR-aur1PR3-papR2-redD*	pRM4-SARPs/pEM2 [7]
Gamma butyrolactone synthases	*scbA-afsA*	pRM4-GBLs
Global regulators	*afsR-adpA-crp-absB-dasR*	pRM4-GRs

**Table 2 molecules-26-06580-t002:** ^1^H NMR (500 MHz in DMSO-*d6*) and ^13^C NMR (125 MHz, DMSO-*d6*) data for **1**.

Position No.	***δ****_C_*_,_ Mult	***δ****_H_* (Mult *, *J* in Hz)
1	95.6, C	
3	69.8, CH	4.74, d (5.0)
4	81.9, C	
4a	141.2, C	
5	181.3, C	
5a	131.4, C	
6	118.9, CH	7.60, dd (7.4, 1.5)
7	137.4, CH	7.82, dd (8.6, 7.4)
8	125.3, CH	7.43, dd (8.6, 1.5)
9	160.8, C	
9-O*H*		11.72, br s
9a	114.9, C	
10	186.3, C	
10a	140.2, C	
11	34.4, CH_2_	3.61, dd (18.1, 5.0)/2.59, d (18.1)
12	174.1, C	
3′	70.9, CH	4.78 ^a^, d (4.2)
4′	65.1 ^a^, CH	5.49, q (3.3)
4′-O*C*OCH_3_	169.9, C	
4′-OCO*CH_3_*	20.8, CH_3_	2.01, s
5′	35.5, CH_2_	H-5′eq:1.89, m/H-5′ax: 1.84, m
6′	62.8, CH	4.22, m
7′	20.2, CH_3_	1.18, d (6.3)
1″	93,7, CH	4.76 ^a^, br s
2″	28.7, CH_2_	1.53, m; 1.40, m
3″	14.3	1.61, m; 1.38, m
4″	64.4	2.57 ^a^, m
4″-N(*CH_3_*)_2_	40.4, 2 × CH_3_	2.54 ^a^, s
5″	65.1 ^a^, CH	3.19, m
6″	17.3, CH_3_	0.47, d (6.2)
1″′	171.6, C	
2″′	51.5, CH	4.37, ddd (8.5, 8.2, 5.0)
3″′	32.6, CH_2_	3.33, dd (13.5, 5.0)/2.94, dd (13.5, 8.5)
2″′-NH		8.22, d (8.2)
2″′-NH*C*OCH_3_	169.3, C	
2″′-NHCO*CH_3_*	22.1, CH_3_	1.67, s

^a^ Overlapping with other ^13^C/^1^H NMR signals. * No multiplicity is specified (m) for ^1^H-NMR signals for which coupling constants (*Js*) could not be measured directly in ^1^H or JRES spectra nor determined from HSQC traces. All assignments were supported by TOCSY, HSQC, and HMBC.

**Table 3 molecules-26-06580-t003:** BLASTN of type II PKS genes from *S. griseus* K-63 (GenBank entry X77865.1) against the genome of *Streptomyces* sp. CA-256286, two hits are displayed for each of the genes.

Gene Name	Origin of the Gene	Best BLAST Hit Score	Pairwise Identity, %	E Value	Position, nt	BGC and Gene Locus Tag
Ketosynthase CDS from X77865.1: CAA54858.1	*S. griseus* K-63	1061.67 682.96	78.8% 73.4%	0 0	3,472,656–3,471,404 6,876,549–6,875,389	BGC 1.20 FBHECJPB_03028 BGC 1.31 FBHECJPB_06072
Putative chain length determination factor CDS from X77865.1: CAA54859.1	*S. griseus* K-63	590.988 365.567	72.4% 69.1%	1.17 × 10^168^ 8.44 × 10^101^	3,471,265–3,470,151 6,875,245–6,874,135	BGC 1.20 FBHECJPB_03027 BGC 1.31 FBHECJPB_06071
Acyl carrier protein CDS from X77865.1: CAA54860.1	*S. griseus* K-63	206.87 62.6008	82.4% 68.8%	1.01 × 10^53^ 2.72 × 10^10^	3,470,076–3,469,873 6,874,057–6,873,904	BGC 1.20 FBHECJPB_03026 BGC 1.31 FBHECJPB_06070
Ketoreductase CDS from X77865.1: CAA54861.1	*S. griseus* K-63	668.533 780.342	79.2% 82.6%	0 0	6,893,267–6,892,495 3,469,673–3,468,908	BGC 1.31 FBHECJPB_06088 BGC 1.20 FBHECJPB_03025
Putative cyclase/dehydratase CDS from X77865.1: CAA54862.1	*S. griseus* K-63	746.078 625.252	78.2% 75.4%	0 4.37 × 10^179^	3,468,839–3,467,912 6,892,412–6,891,493	BGC 1.20 FBHECJPB_03024 BGC 1.31 FBHECJPB_06087

**Table 4 molecules-26-06580-t004:** BLASTN analysis of *S. coelicolor* mycothiol detoxification genes against the genome of *Streptomyces* sp. CA-256286 to identify homologs.

Gene Name	Origin of the Gene	Best BLAST Hit Score	Pairwise Identity, %	E Value	Position, nt	BGC and Locus Tags
*mshA* (SCO4204)	*S. coelicolor* A3(2)	1296.62	85.2%	0	3,445,621–3,444,352 No other good hits	BGC 1.20 FBHECJPB_03001
*mshB* (SCO5126)	*S. coelicolor* A3(2)	431.39 75.224	76.5% 73.4%	9.22 × 10^121^ 1.51 × 10^13^	2,457,013–2,456,142 2,642,174–2,642,051	Close to BGC 1.18 FBHECJPB_02090 Close to BGC 1.18 FBHECJPB_02245
*mshC* (SCO1663)	*S. coelicolor* A3(2)	1362.1 118.505	86.7% 70.4%	0 1.98 × 10^26^	6,462,580–6,463,808 3,428,519–3,428,260	Not part of any BGC FBHECJPB_05696 Close to BGC 1.20 FBHECJPB_02982
*mshD* (SCO4151)	*S. coelicolor* A3(2)	780.342	79.8%	0	3,532,239–3,533,119 No other good hits	Close to BGC 1.20 FBHECJPB_03096
Mca (SCO4967)	*S. coelicolor* A3(2)	1126.59	88.3%	0	2,642,188–2,641,307	Close to BGC 1.18 FBHECJPB_02245

**Table 5 molecules-26-06580-t005:** BLAST (tblastn) against the genome of CA-256286 using five proteins (SpnO, SpnN, SpnQ, SpnR and SpnS proteins) from *Saccharopolyspora spinosa* NRRL 18537 spinosyn pathway. If there were several results, only the top 2 were recorded.

Protein Name	Origin of the Protein	Best BLAST Hit Score	Pairwise Identity, %	E Value	Position, nt	BGC and Locus Tag
SpnR	*Saccharopolyspora spinosa* NRRL 18537	503.056 168.318	63.4% 33.4%	1.43 × 10^160^ 3.58 × 10^45^	6,893,551–6,894,681 5,334,768–5,335,814	BGC 1.31 FBHECJPB_06089 Not part of any BGC
SpnQ	*Saccharopolyspora spinosa* NRRL 18537	625.165 141.739	68.6% 30.3%	0 6.48 × 10^36^	6,881,515–6,882,825 5,334,753–5,335,826	BGC 1.31 FBHECJPB_06078 Not part of any BGC
SpnO	*Saccharopolyspora spinosa* NRRL 18537	409.068	48.2%	2.11 × 10^126^	6,879,120–6,880,454	BGC 1.31 FBHECJPB_06076
SpnN	*Saccharopolyspora spinosa* NRRL 18537	265.003	50.8%	7.04 × 10^79^	6,880,460–6,881,404	BGC 1. 31 FBHECJPB_06077
SpnS	*Saccharopolyspora spinosa* NRRL 18537	219.164	46.5%	3.11 × 10^64^	6,871,449–6,870,763	BGC 1.31 FBHECJPB_06067

## Data Availability

MS raw data has been deposited at GNPS (https://gnps.ucsd.edu/, accessed on 1 October 2021) [62] under the MassIVE ID number: MSV000087475.

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
