# Peer review of "Activation and Identification of a Griseusin Cluster in Streptomyces sp. CA-256286 by Employing Transcriptional Regulators and Multi-Omics Methods"

_molecules, 2021, doi:10.3390/molecules26216580_

Round 1
Reviewer 1 Report
This manuscript reported the natural products discovery through heterologously expressing transcriptional regulators in Streptomyces sp. CA-256286, which lead to the isolation of one new compound, that believed to be the inactivated form of antimicrobial metabolite, griseusin. In addition, the putative gene cluster has identified.
The planar structure is very hard to determine, HMBC could not confirm the whole connection due to a large number of quaternary carbons, but with existing structure of 3 give important clue of the core structure of 1 and 2. The structure is very unique, how the cysteine attach to the structure is very interesting. Regioselectivity of N-acetylation and N-methylation is also fascinating.
Author Response
We appreciate the reviewer´s comment. Indeed, the planar structure was very difficult to determine due to the presence of so many quaternary carbons in the core structure. Fortunately, some key HMBC correlations and some characteristic chemical shifts for carbons within the pyranonaphtoquinone core were very helpful. Of course, the existence of the compound 3 greatly facilitated this task, although the attachment of the N-AcCys moiety to C-4 in the structure of 2 complicated the situation in turn.
Reviewer 2 Report
The article entitled “Activation and identification of a griseusin cluster in Streptomyces sp. CA-256286 by employing transcriptional regulators and multi-omics methods” demonstrated a workflow to activate the silent BGCs and the subsequent approaches of identification and characterization. In general, the manuscript was well written, and the results were presented clearly. But a revision is required before recommending it for publication.
- In the introduction section, suggest adding the objective of this study.
- The resolution of figure 3 should be improved.
- In figure 4, suggest using different symbols rather than colors.
- The authors should correct the typo error in the legend of figure 5, (i)
- Do not feel comfortable with the resolution of figure 6. The X-axis is not readable.
- In section 4.8, the sentences regarding the sample collection can be moved to section 4.1. and the authors can just mention the samples are both for RNA sequencing and proteomics.
- The authors should briefly introduce the approach or methodology of transcriptomics analysis and reference was required here.
Author Response
We addressed the reviewer's comment as follows:
- In the introduction section, suggest adding the objective of this study.
- An objective was added in the end of the introduction section.
- The resolution of figure 3 should be improved.
- Figure 1, 2 and 3 were inserted in higher resolution.
- In figure 4, suggest using different symbols rather than colors.
- Changed to triangles and squares, the info is also added in the figure legend.
- The authors should correct the typo error in the legend of figure 5, (i)
- (i) was changed to (ii). Thank you for spotting this.
- Do not feel comfortable with the resolution of figure 6. The X-axis is not readable.
- Figure 6 was inserted in higher resolution.
- In section 4.8, the sentences regarding the sample collection can be moved to section 4.1. and the authors can just mention the samples are both for RNA sequencing and proteomics.
- The information was moved to section 4.1.
- The authors should briefly introduce the approach or methodology of transcriptomics analysis and reference was required here.
- Information is added in the beginning of section 2.5 and a reference was included. Note that the update in reference number caused an update in all references with the track-changes function.